# Learning to correct spectral methods for simulating turbulent flows

**Gideon Dresdner**                                                    *gideond@gmail.com*
*Google Research and ETH Zurich, Department for Computer Science*

**Dmitrii Kochkov**                                                    *dkochkov@google.com*
*Google Research*

**Peter Norgaard**                                                     *pnorgaard@google.com*
*Google Research*

**Leonardo Zepeda-Núñez**                                             *lzepedanunez@google.com*
*Google Research*

**Jamie A. Smith**                                                     *jamieas@google.com*
*Google Research*

**Michael P. Brenner**                                                 *mbrenner@google.com*
*Google Research*

**Stephan Hoyer**                                                      *shoyer@google.com*
*Google Research*

## Abstract

Despite their ubiquity throughout science and engineering, only a handful of partial differential equations (PDEs) have analytical, or closed-form solutions. This motivates a vast amount of classical work on numerical simulation of PDEs and more recently, a whirlwind of research into data-driven techniques leveraging machine learning (ML). A recent line of work indicates that a hybrid of classical numerical techniques and machine learning can offer significant improvements over either approach alone. In this work, we show that the choice of the numerical scheme is crucial when incorporating physics-based priors. We build upon Fourier-based spectral methods, which are known to be more efficient than other numerical schemes for simulating PDEs with smooth and periodic solutions. Specifically, we develop ML-augmented spectral solvers for three common PDEs of fluid dynamics. Our models are more accurate $(2 - 4\times)$ than standard spectral solvers at the same resolution but have longer overall runtimes $(\sim 2\times)$, due to the additional runtime cost of the neural network component. We also demonstrate a handful of key design principles for combining machine learning and numerical methods for solving PDEs.

## 1 Introduction

The numerical simulation of nonlinear partial differential equations (PDEs) permeates science and engineering, from the prediction of weather and climate to the design of engineering systems. Unfortunately, solving PDEs on the fine grids required for high-fidelity simulations is often infeasible due to its prohibitive computational cost. This leads to inevitable trade-offs between runtime and accuracy. The status quo is to solve PDEs on grids that are coarse enough to be computationally feasible but are often too coarse to resolve all phenomena of interest. One classical approach is to derive coarse-grained surrogate PDEs such as Reynold's Averaged Navier Stokes (RANS) and Large Eddy Simulation (LES) (Pope, 2000), which in principle can be

accurately solved on coarse grids. This family of approaches has enjoyed widespread success but is tedious to perform, PDE-specific, and suffers from inherent accuracy limitations (Durbin, 2018; Pope, 2004).

Machine learning (ML) has the potential to overcome many of these limitations by inferring coarse-grained models from high-resolution ground-truth simulation data. Turbulent fluid flow is an application domain that has already reaped some of these benefits. Pure ML methods have achieved impressive results, in terms of accuracy, on a diverse set of fluid flow problems (Li et al., 2021; Stachenfeld et al., 2022). Going beyond accuracy, hybrid methods have combined classical numerical simulation with ML to improve stability and generalize to new physical systems, e.g., out of sample distributions with different forcing setups (Kochkov et al., 2021; List et al., 2022).

However, hybrid methods have been limited to low-order finite-difference and finite volume methods, with the exception of one recent study (Frezat et al., 2022). Beyond finite differences and finite volumes, there is a broad field of established numerical methods for solving PDEs. In this paper, we focus on spectral methods which are used throughout computational physics (Trefethen, 2000; Burns et al., 2020) and constitute the core of the state-of-the-art weather forecasting system (Roberts et al., 2018). Spectral methods, when applicable, are often preferred over other numerical methods because they can be more accurate for equations with smooth solutions. In fact, their accuracy rivals that of the recent progress made by ML. This begs the question: *Can we improve spectral solvers of turbulent fluid flows using learned corrections of coarse-grained simulations?*

Our contributions are as follows:

1. We propose a hybrid physics machine learning method that provides sub-grid corrections to classical spectral methods.

2. We explore two toy 1D problems: the unstable Burgers' equation and the Kuramoto-Sivashinsky the hybrid model is able to make the largest improvements.

3. We compare spectral, finite-volume, ML-only, and our hybrid model on a 2D forced turbulence task. Our hybrid models provide some improvement on the accuracy of spectral-only methods, which themselves perform remarkably well compared to recently proposed ML methods. Furthermore, our results show that a key modeling choice for both hybrid and pure ML models is the use of velocity- rather than vorticity-based state representations.

## 1.1 Related work

The study of turbulent fluid dynamics is vast. We refer to Pope (2000) for a thorough introduction. Classical approaches tend to derive mathematical approximations to the governing equations in an *a priori* manner. Recently, there has been an explosion of work in data-driven methods at the interface of computational fluid dynamics (CFD) and machine learning (ML). We loosely organize this recent work into three main categories:

**Purely Learned Surrogates** fully replace numerical schemes from CFD with purely learned surrogate models. A number of different architectures have been explored, including multi-scale convolutional neural networks (Ronneberger et al., 2015; Wang et al., 2020), graph neural networks (Sanchez-Gonzalez et al., 2020), and Encoder-Process-Decoder architectures (Stachenfeld et al., 2022).

**Operator Learning** seeks to learn the differential operator directly by mimicking the analytical properties of its class, such as pseudo-differential or Fourier integral operators but without explicit physics-informed components. These methods often leverage the Fourier transform (Li et al., 2021; Tran et al., 2021) and the off-diagonal low-rank structure of the associated Green's function (Fan et al., 2019; Li et al., 2020).

**Hybrid Physics-ML** is a recent line of work which seeks to combine classical numerical methods with contemporary data-driven deep learning techniques (Mishra, 2018; Bar-Sinai et al., 2019; Kochkov et al., 2021; List et al., 2022; Frezat et al., 2022; Bruno et al., 2021) These approaches seek to *learn* corrections to low-resolution numerical schemes using high-resolution simulation data. The goal is to combine the best of both worlds — the simplicity of PDE-based governing equations, with the expressive power of neural

networks. In their recent work, Frezat et al. (2022) augmented spectral solvers for closure models related to climate modeling. Working in this vein, we also learn corrections to spectral methods. In contrast, rather than focusing on climate models, we provide key ML architecture choices and include comparisons to classical numerical methods on multiple grid resolutions.

## 2 Spectral methods for fluids

Spectral methods are a powerful method for finding high-accuracy solutions to PDEs and are often the method of choice for solving smooth PDEs with simple boundary conditions and geometries. There is an extensive literature on the theoretical and practical underpinnings of spectral methods (Gottlieb & Orszag, 1977; Gottlieb & Hesthaven, 2001; Canuto et al., 2007; Kopriva, 2009), particularly for methods based on Fourier spectral collocation (Trefethen, 2000; Boyd, 2001). Below, we provide a succinct introduction .

### 2.1 Partial differential equations for turbulent fluids

Let $\mathbf{u} : \mathbb{R}^d \times \mathbb{R}^+ \to \mathbb{R}^{d'}$ be a time-varying vector field, for dimensions $d$ and $d'$. We study PDEs of the form

$$\partial_t \mathbf{u} = D\mathbf{u} + N(\mathbf{u}), \tag{1}$$

plus initial and boundary conditions. Here $D$ is a linear partial differential operator, and $N$ is a non-linear term. Equations of this form dictate the temporal evolution of $\mathbf{u}$ driven by its variation in space. In practice, PDEs are solved by discretizing in space and time, which converts the continuous PDE into a set of update rules for vectors of coefficients to approximate the state $\mathbf{u}$ in some discrete basis, e.g., on a grid. For time-dependent PDEs, temporal resolution must be scaled proportionally to spatial resolution to maintain an accurate solution. Thus, runtime for PDE solvers is $O(n^{d+1})$, where $d$ is the number of spatial dimensions and $n$ is the number of discretization points per dimension.

For simulations of fluids, the differential operator is typically either diffusive, $D = \partial_x^2$, or hyper-diffusive, $D = \partial_x^4$. And, the non-linearity is a convective term, $N = \frac{1}{2}\partial_x(\mathbf{u}^2) = \mathbf{u}\,\partial_x\mathbf{u}$. *Diffusion* is the tendency of the fluid velocity to become uniform due to internal friction, and *convection* is the tendency of the fluid to be transported by its own inertia. Turbulent flows are characterized by a convective term that is much stronger than diffusion.

For most turbulent flows of interest, closely approximating the exact PDE (known as Direct Numerical Simulation) is computationally intractable because it requires prohibitively high grid resolution. Instead, coarse-grained approximations of the PDE are solved, known as "Large Eddy Simulation" (LES). LES augments Equation (1) by adding a correction term determined by a closure model to account for averaged effects over fine spatial length scales. In practice, this term is often omitted due to the difficulty of deriving appropriate closure formulas ("implicit LES") and the PDE is simply simulated with the finest computationally feasible grid resolution. In our case, correction terms are an opportunity for machine learning. If we can accurately model PDEs on coarser grids with suitable correction terms, we may be able to significantly reduce the computational cost of large-scale simulations.

### 2.2 The appeal of the Fourier basis for modeling PDEs

Let us further assume that $\mathbf{u}(x,t) : [0, 2\pi] \times \mathbb{R}^+ \to \mathbb{R}$ in Equation (1) is $2\pi$-periodic, square-integrable for all times $t$, and for simplicity, one-dimensional. Consider the Fourier coefficients $\widehat{\mathbf{u}}^t$ of $\mathbf{u}(x,t)$, truncated to lowest $K+1$ frequencies (for even $K$):

$$\widehat{\mathbf{u}}^t = (\widehat{\mathbf{u}}^t_{-K/2}, \ldots, \widehat{\mathbf{u}}^t_k, \ldots, \widehat{\mathbf{u}}^t_{K/2}) \quad \text{where} \quad \widehat{\mathbf{u}}^t_k = \frac{1}{2\pi}\int_0^{2\pi} \mathbf{u}(x,t)e^{-ik\cdot x}dx. \tag{2}$$

$\widehat{\mathbf{u}}^t$ is a vector representing the solution at time $t$, containing the coefficients of the Fourier basis $e^{ikx}$ for $k \in \{-K/2, \ldots, K/2\}$. In general, the integral $\widehat{\mathbf{u}}^t_k$ has no analytical solution and so we approximate it using a trapezoidal quadrature on $K+1$ points. Namely, we approximate it by sampling $\mathbf{u}(x,t)$ on an equispaced grid. The Fast Fourier Transform (FFT) (Cooley & Tukey, 1965) computes these Fourier coefficients efficiently in

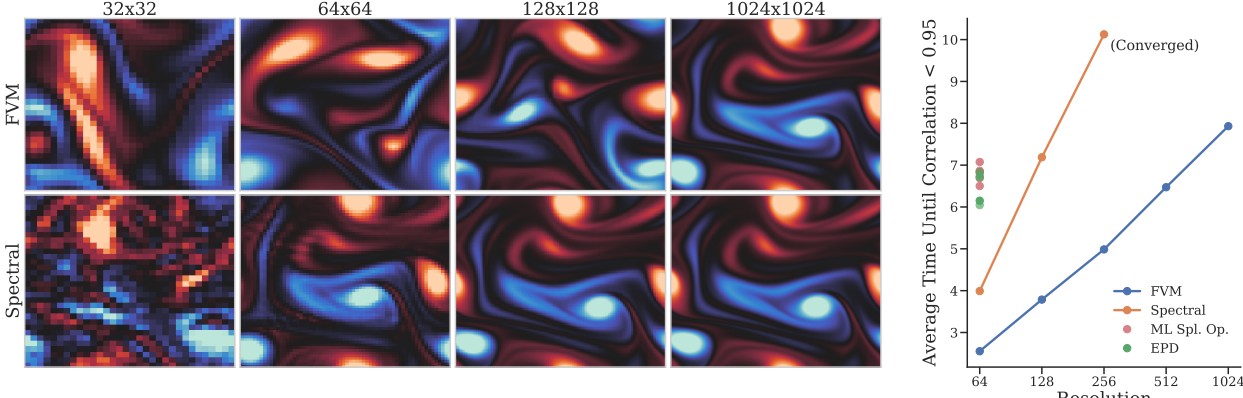

Figure 1: Comparing finite volume (FVM) to spectral convergence for 2D turbulence. *Left*: Vorticity fields evolved up to time 9.0 with different grid sizes and numerical methods, starting from an identical initial condition. Qualitatively, it is clear that at resolution 64x64, the spectral method has already captured most of features of the high-resolution state. The FVM looks sharp, but clearly differs. At sufficiently high resolution, the methods converge to the same solution. *Right*: Here we compare each method with high-resolution 2048x2048 ground truth. We plot the time until the first dip below 0.95 of the correlation with the ground truth. Note that the initial conditions for the finite volume and the spectral method are sampled from the same distribution, but are not identical. At resolution 256, the spectral method has converged within the accuracy limitations of single precision, so we omit higher resolutions. For the EPD baseline and our ML Split Operator (ML Spl. Op.) models, we show performance across five different neural network parameter initializations.

log-linear time. Spectral methods for PDEs leverage the fact that differentiation in the Fourier domain can be calculated by element-wise multiplication according to the identity $\partial_x \widehat{\mathbf{u}}_k = ik\widehat{\mathbf{u}}_k$. This, in turn, makes inverting linear differential operators easy since it is simply element-wise division (Trefethen, 2000).

When time-dependent PDEs include non-linear terms, spectral methods evaluate these terms on a grid in real-space, which requires forward and inverse FFTs inside each time-step. These transforms introduce two sources of error. First, the quadrature rules for $\widehat{\mathbf{u}}_k^t$ produce only an approximation to the integral. Second, there will be a truncation error when the number of frequencies $K + 1$ is less than the bandwidth of $\mathbf{u}(x, t)$. Remarkably, it is a well-known fact of Fourier analysis that both of these errors decay super-algebraically (i.e., faster than any power of $1/K$) if $\mathbf{u}(x, t)$ is periodic and is in $C^\infty$ (Trefethen, 2000).[1] Thus, relatively few discretization points are needed to represent solutions which are $C^\infty$.

Because of these favorable convergence properties, spectral methods often outshine their finite difference counterparts. For example, spectral methods are used for large-scale simulations of turbulence on GPU super-computers (Yeung & Ravikumar, 2020). See Figure 1 for a simple comparison in the case of 2D turbulence. This motivates us to start from spectral methods, rather than finite difference methods, and further improve them using machine learning.

## 3 Learned split operators for correcting spectral methods

Similarly to classical spectral numerical methods, we solve Equation (1) by representing our state in a finite Fourier basis as $\widehat{\mathbf{u}}^t$ and integrating forward in time. We model the fully known differential operators $D$ and $N$ using a standard spectral method, denoted Physics-Step. The machine learning component, denoted Learned-Correction$(\,\cdot\,; \theta)$, contains tune-able parameters $\theta$ which are optimized during training to match high-resolution simulations.

---

[1]Convergence is exponential for analytic $\mathbf{u}$ (Tadmor, 1986).

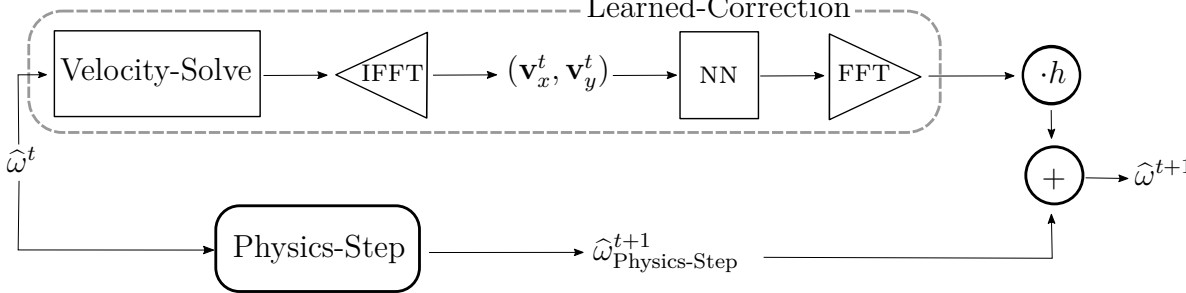

Figure 2: Diagram summarizing our model described in Equation (3) for the 2D Navier-Stokes equation. The input, vorticity $\widehat{\omega}^t$, is processed by two independent components—Learned-Correction and Physics-Step—operating at different time-scales. The output of Learned-Correction is weighted by $h$, as in a basic first-order Euler stepping scheme, and combined with the output of the Physics-Step to give the state at the next time step.

Building upon traditional physics-based solvers, we use a simple explicit-Euler time integrator for the correction term. This yields the following update equation:

$$\widehat{\mathbf{u}}^{t+1} = \text{Physics-Step}(\widehat{\mathbf{u}}^t) + h \cdot \text{Learned-Correction}(\widehat{\mathbf{u}}^t; \theta), \tag{3}$$

where $h \in \mathbb{R}$ is the time step size. Figure 2 presents a diagram of Equation (3) applied to the Navier-Stokes equation.

## 3.1 Convolutional layers for encoding the physical prior of locality

For a small time step $h$, the solution at time $t + h$ only depends locally in space on the solution at time $t$ (within a dependency cone usually encoded by the Courant–Friedrichs–Lewy (CFL) condition). Following previous work (see Sec. 1.1, Hybrid Physics-ML), we incorporate this assumption into the machine learning component of our model using Convolutional Neural Networks (CNN or ConvNet) which, by design, only learn local features. We now provide a high-level view of our modeling choices:

**Real-space vs. frequency-space.** Since the Fourier basis is global, each coefficient $\widehat{\mathbf{u}}_k^t$ contains information from the full spatial domain as shown in Equation (2). Thus, in order to maintain spatial features, which are local, we apply the ConvNet component of our Neural Split Operator model in real-space. This is accomplished efficiently via inverse-FFT and FFT to map the signal back and forth between frequency- and real-space.

**ConvNet Padding.** In this work, we tackle problems with periodic boundary conditions. This makes periodic padding a natural choice for the ConvNets.

**Neural architecture.** For both 1D and 2D problems we used an Encoder-Process-Decoder (EPD) architecture (Stachenfeld et al., 2022). This facilitates easier comparison to pure neural network baselines. See Section 4.2 for a detailed description of EPD models and Appendix C for further information.

## 3.2 The split operator method for combining time scales

Due to a variety of considerations — numerical stability, computational feasibility, etc. — each term of a PDE often warrants its own time-advancing method. This motivates *split operator methods* (Strang, 1968; McLachlan & Quispel, 2002), a popular tool for solving PDEs which combine different time integrators. In this work, we use the split operator approach to incorporate the additional terms given by the neural network.

In the usual pattern of spectral methods, Physics-Step itself is split into two components corresponding to the $D$ and $N$ terms of Equation (1). The $D$ term of Physics-Step is solved with a Crank-Nicolson method

and the $N$ part is solved using explicit 4th-order Runge-Kutta, which effectively runs at a time-step of $h/4$. The Learned-Correction component is solved using a vanilla first-order Euler time-step with step size $h$, which when compared to incorporating the learned component in the 4th-order Runge-Kutta solver, is more accurate and has 4x faster runtime (see Sec. 4). Alternatively, omitting Physics-Step from Equation (3) gives a Neural ODE model (Chen et al., 2018) with first-order time-stepping which is precisely the EPD model described by Stachenfeld et al. (2022). This model serves as a strong baseline as shown in Section 4.

### 3.3 Physics-based solvers for calculating neural-net inputs

A final important consideration is the choice of input representation for the machine learning model. We found that ML models operate better in velocity-space whereas vorticity is the more suitable representation for the numerical solver. We were able to improve accuracy by incorporating a physics-based data pre-processing step, e.g., a velocity-solve operation, for the inputs of the neural network component. Our overall model is Learned-Correction = FFT(NN(IFFT(State-Transform($\widehat{\mathbf{u}}^t$)))), where NN is implemented a periodic ConvNet. For our 1D test problems, State-Transform is the identity transformation, but for 2D Navier-Stokes (depicted in Fig. 2) we perform a velocity-solve operation to calculate velocity from vorticity. This turns out to be key modeling choice, as described in Section 4.

### 3.4 Training and evaluation

**Data preparation for spectral solvers.** For ground truth data, we use fully resolved simulations, which we then downsample to the coarse target resolution. Choosing the downsampling procedure is a key decision for coarse-grained solvers (Frezat et al., 2022). In this work, the fully resolved trajectories are first truncated to the target wavenumber (i.e., an ideal low-pass filter). Then, we apply an exponential filter of the form $\tilde{\mathbf{u}}_k = \exp(-\alpha|k/k_{\max}|^{2p})\widehat{\mathbf{u}}_k$, where $\tilde{\mathbf{u}}_k$ denotes the filtered field and $k$ is the $k$-th wavenumber (Canuto et al., 2006). We obtained stable trajectories using a relatively weak filter with $\alpha = 6$ and $p = 16$. The exponential filter smooths discontinuities in the PDE solution, which otherwise manifest themselves globally in real space as ringing artifacts known as Gibbs Phenomena (Canuto et al., 2006).

Filtering is also used to correct aliasing errors in spectral methods that arise when evaluating non-linear terms (Gottlieb & Hesthaven, 2001). Ideally, filtering for aliasing errors would be spatially adaptive (Boyd, 1996). In practice, however, filters for both aliasing and truncation are often chosen heuristically. While one might aspire to learn these heuristics from data, our attempts at doing so were unsuccessful. In part this is because insufficiently filtered simulations will often entirely diverge rather than accumulate small errors. Thus, in addition to filtering the downsampled training data, we also used the exponential filter on the outputs of Physics-Step of Equation (3) and consider this to be another component of Physics-Step. Omitting this filtering also resulted in global errors which were impossible for the learned component to correct. Figure 3 summarizes our data generation pipeline, including an explicit filtering step. In Section 4, we present a failure mode on a model without filtering (see Fig. 5).

**Training loss.** We train our models to minimize the squared-error over some number of unrolled prediction steps, which allows our model to account for compounding errors (Um et al., 2020). Let $\overline{\mathbf{u}}(x, t)$ denote our prediction at time $t$, then our training objective is given by $\beta \sum_{x, t \le T} |\overline{\mathbf{u}}(x, t) - \mathbf{u}(x, t)|^2$. The scaling constant $\beta$ is chosen so that the loss of predicting "no change" is one, i.e., $\beta^{-1} = \sum_{x, t \le T} |\mathbf{u}(x, 0) - \mathbf{u}(x, t)|^2$. Relative to the thousands of time-steps over which we hope to simulate accurately, we unroll over a relatively small number during training (e.g., $T = 32$ for 2D turbulence) because training over long time windows is less efficient and less stable (Kochkov et al., 2021).

**Measuring convergence.** We seek to optimize the accuracy of coarse-grained simulations. More specifically, we aim to make a coarse resolution simulation as similar as possible to a high-resolution simulation which has been coarse-grained in post-processing. Following Kochkov et al. (2021), we measure convergence to a fully resolved, high-resolution ground-truth at each time $t$ in terms of mean absolute error (MAE), correlation, and time until correlation is less than 0.95. MAE is defined as $\sum_x |\overline{\mathbf{u}}(x, t) - \mathbf{u}(x, t)|$ and correlation is defined $\mathrm{Corr}[\mathbf{u}(\cdot, t), \overline{\mathbf{u}}(\cdot, t)] = \sum_x (\overline{\mathbf{u}}(x, t) \cdot \mathbf{u}(x, t))/(\|\overline{\mathbf{u}}(x, t)\|_2 \|\mathbf{u}(x, t)\|_2)$ (since our flows have mean zero). Finally, we compute the first time step in which the correlation dips below 0.95, i.e.

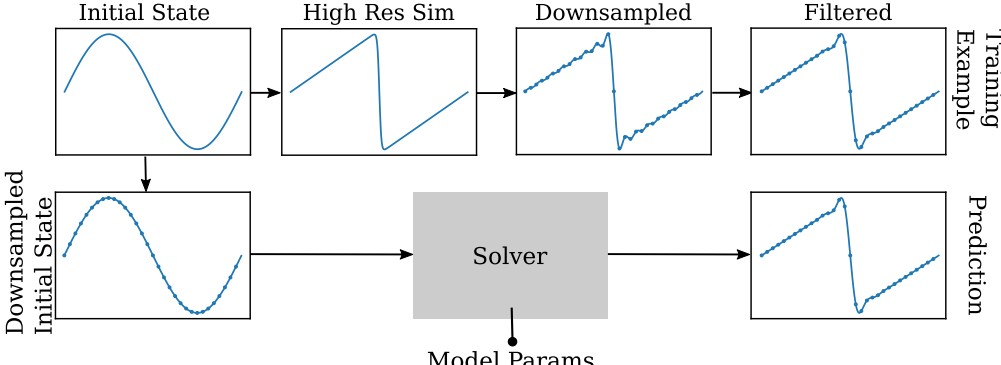

Figure 3: Diagram of our training pipeline. Starting with a high resolution initial state, we run it forward using a high-resolution spectral solver. Then we downsample by truncating higher frequencies in the Fourier representation. This can cause "ringing effects" for which the standard approach is to apply a filtering operator. Finally, we train a solver to mimic this process as closely as possible, as measured in $\ell_2$-loss.

$\min\{t \mid \mathrm{Corr}[\mathbf{u}(\cdot, t), \overline{\mathbf{u}}(\cdot, t)] < 0.95\}$. While MAE is precise up to floating point precision, it is not readily interpretable. Whereas, correlation is less sensitive but more interpretable. For this reason, we prefer to report correlation. This potential redundancy is demonstrated in our experiment on the KS equation (Sec. 4.3), where we included both MAE and correlation.

For the 2D Navier-Stokes equation, we only report correlation because it is sufficient. For the unstable Burgers' equation, the situation was the opposite: We only reported MAE because measuring correlation did not provide additional insight — all models had correlation values close to 1.0, whereas MAE was sensitive to the improved performance of our method.

## 4 Results

### 4.1 Model equations

We showcase our method using three model equations which capture many of the algorithmic difficulties present in more complex systems: 1D Kuramoto–Sivashinsky (KS) equation, 1D unstable Burgers' equation, and 2D Kolmogorov flow, a variant of incompressible Navier-Stokes with constant forcing.

The KS equation has smooth solutions which spectral methods are well-designed to solve. Therefore, it is not surprising that, while our method does improve over spectral-only methods, that improvement is not significant. On the other hand, the unstable Burgers' equation presents a test case in which classical spectral methods struggle near discontinuities. Here, our method outperforms spectral-only methods. Finally, with two-dimensional Kolmogorov flow, we demonstrate our method on a more challenging fluid simulation. Kolmogorov flow exhibits multiscale behavior in addition to smooth, chaotic dynamics. Without any modification, spectral-only methods are already competitive with the latest hybrid methods (Kochkov et al., 2021). Similar to the 1D KS equation, our method is able to provide some improvement in this case.

### 4.2 Baselines

Our hybrid spectral-ML method naturally gives rise to two types of baselines: spectral-only and ML-only. On all three equations, we compare to spectral-only at various resolutions, e.g., "Spectral 32" refers to the spectral-only method with 32 grid points.

For Kolmogorov flow, we compare to two additional baselines: (1) the Encoder-Process-Decoder (EPD) model (Stachenfeld et al., 2022), a state-of-the-art ML-only model, and (2) a finite volume method (FVM) as implemented by Kochkov et al. (2021) — a standard numerical technique which serves as an alternative to the spectral method.

The EPD architecture acts in three steps. First, the spatial state is embedded into a high-dimensional space using a feed-forward neural network encoder architecture. Then, either another feed-forward or a recurrent architecture is applied to process the embedded state. Finally, the output of the process step is projected to back to the spatial state using another feed-forward decoder module. This method has recently achieved state-of-the-art on a wide range of one-, two-, and three-dimensional problems. For consistency, we used an identical EPD model for the learned component of our model (Eq. (3)).

Finally, we included a second-order FVM on a staggered grid to illustrate the strength of the spectral baselines. This gives context to compare to previous work in hybrid methods (Bar-Sinai et al., 2019; Kochkov et al., 2021) which use this FVM model as a baseline.

We avoided extensive comparisons to classical subgrid modeling, such as Large Eddy Simulation (LES), since LES is itself a nuanced class of methods with many tunable parameters. Instead, our physics-only baselines are implicit LES models — a widely-used and well-understood model class which serves as a consistent, parameter-free baseline. In contrast, explicit sub-grid-scale models such as Smagorinsky models include tunable parameters, with the optimal choice depending on the scenario of interest. Furthermore, explicit LES models typically focus on matching the energy spectrum rather than minimizing point-wise errors (List et al., 2022).

### 4.3 Kuramoto-Sivashinsky (KS) equation

The Kuramoto-Sivashinsky (KS) equation is a model of unstable flame fronts. In the PDE literature, it is a popular model system because it exhibits chaotic dynamics in only a single dimension. The KS equation is defined as

$$\partial_t \mathbf{u} = -\mathbf{u}\partial_x \mathbf{u} - \partial_x^4 \mathbf{u} - \partial_x^2 \mathbf{u}. \tag{4}$$

The three terms on the right hand side of this equation correspond to convection, hyper-diffusion and anti-diffusion, which drives the system away from equilibrium.

Because solutions are smooth, spectral methods are able to capture the dynamics of the KS equation extremely well. At a resolution of 64, the simulation is already effectively converged to the limits of single precision arithmetic, i.e., it enjoys a perfect correlation with a high-resolution ground truth.

Considering how well-suited spectral methods are for modeling this equation, it is remarkable that our ML-Physics hybrid model is able to achieve any improvement at all. Looking qualitatively at the left panel of Figure 4, there is a clear improvement over the spectral 32 baseline.

### 4.4 Unstable Burgers' equation

Burgers' equation is a simple one-dimensional non-linear PDE which is used as a toy model of compressible fluid dynamics. The characteristic behavior of Burgers' equation is that it develops shock waves. Practitioners use this equation to test the accuracy of discretization schemes near discontinuities.

Like turbulent flows, the behavior of Burgers' equation is dominated by the convection, but unlike turbulent flows, it is not chaotic. Prior studies of ML applied to Burgers' equation imposed random forcings (Bar-Sinai et al., 2019; Um et al., 2020), but due to the non-choatic nature of the equation, discretization errors decay rather than compounding over time. Instead, we use an unstable viscous Burgers' equation which slightly amplifies low frequencies in order to make the dynamics chaotic. Sakaguchi (1999) provides the following definition:

$$\partial_t \mathbf{u} = -u\partial_x \mathbf{u} + \nu\partial_x^2 \mathbf{u} + \int_0^L g(x - x')\partial_x^2 \mathbf{u}(x')dx' \tag{5}$$

for viscosity $\nu > 0$ and domain size $L$. The three terms in this unstable Burgers' equation correspond to convection, diffusion and a scale-selective amplification of low-frequency signals. The convolution $g(x - x')$ amplifies low frequencies, decaying smoothly to zero as the frequency increases but is best described in the Fourier space, which we defer to Appendix A.

Shock waves are challenging to model using the Fourier basis because there are discontinuities in the solution (Sec. 3.4). Due to the mixing of length scales introduced by convection, these errors can quickly propagate

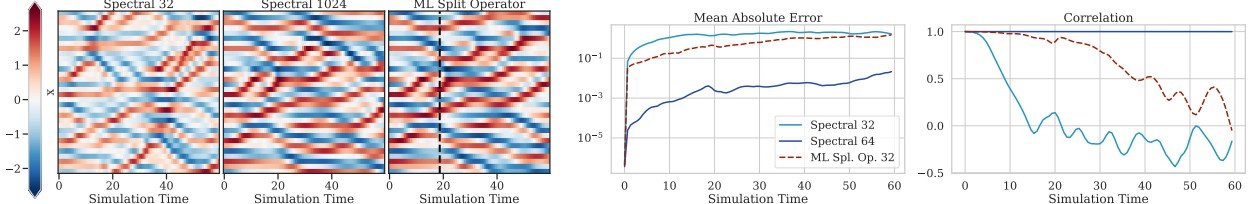

Figure 4: Comparing our model to spectral-only baselines on the KS equation. The spectral method is essentially converged to a 1024 ground truth model at a resolution of 64. Vertical dashed line indicates the first time step in which our model's correlation with the ground truth is less than 0.9. Our model is still able to some improvement over a coarse resolution (Spectral 32) baseline.

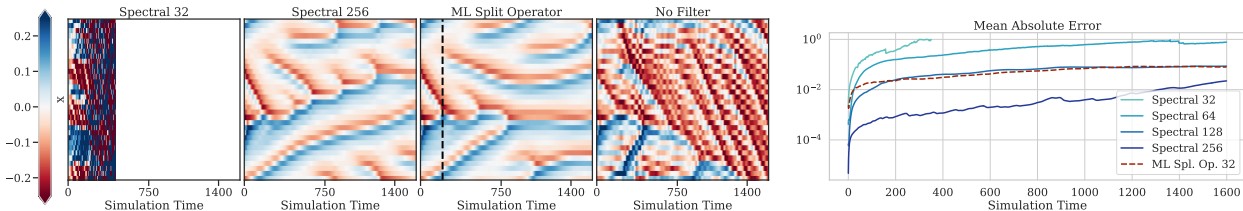

Figure 5: Comparing our model to spectral-only baselines on the unstable Burgers' equation (Eq. (5)). Vertical dashed line indicates the first time step in which our model's correlation with the ground truth is less than 0.9. Not only is our model stable unlike the coarse (Spectral 32) baseline, it also performs on par with a medium-grain (128) baseline, demonstrating a 4x improvement in spatial accuracy. If the Physics-Step component of Equation (3) does not include the exponential filter as described in Section 3.4, the Learned-Correction component diverges completely (*Left* panel, "No Filter").

to affect the overall dynamics. This often results in instability at low resolutions, evidenced in our results (Fig. 5, resolution 32). While there are classical methods for addressing discontinuities, e.g., adding a hyperviscosity term, they often modify the underlying dynamics that they are trying to model. The inadequacy of unaltered spectral methods for solving this problem explains why our method is able to achieve approximately 4x improvement in spatial resolution over a spectral baseline. Results are summarized in Figure 5.

### 4.5 2D turbulence

We consider 2D turbulence described by the incompressible Navier-Stokes equation with Kolmogorov forcing (Kochkov et al., 2021). This equation can be written either in terms of a velocity vector field $\mathbf{v}(x, y) = (\mathbf{v}_x, \mathbf{v}_y)$ or a scalar vorticity field $\omega := \partial_x \mathbf{v}_y - \partial_y \mathbf{v}_x$ (Boffetta & Ecke, 2012). Here we use a vorticity formulation, which is most convenient for spectral methods and avoids the need to separately enforce the incompressibility condition $\nabla \cdot \mathbf{v} = 0$. The equation is given by

$$\partial_t \omega = -\mathbf{v} \cdot \nabla \omega + \nu \nabla^2 \omega - \alpha \omega + f, \tag{6}$$

where the terms correspond to convection, diffusion, linear damping and a constant forcing $f$ (App. B). The velocity vector field can be recovered from the vorticity field by solving a Poisson equation $-\nabla^2 \psi = \omega$ for the stream function $\psi$ and then using the relation $\mathbf{v}(x, y) = (\partial_y \psi, -\partial_x \psi)$. This velocity-solve operation is computed via element-wise multiplication and division in the Fourier basis.

For this example, we compared four types of models: spectral-only, FVM-only, EPD (ML-only), and two types of hybrid methods: our split operator method and a nonlinear term correction method. The ML models are all trained to minimize the error of predicted velocities, which are obtained via a velocity-solve for the spectral-only, EPD and hybrid models that use vorticity as their state representation. The nonlinear term correction method uses a neural network to correct the inputs to the nonlinear term $\mathbf{v} \cdot \nabla \omega$ and avoids using classical correction techniques, e.g., the so-called three-over-two rule (Orszag, 1971).

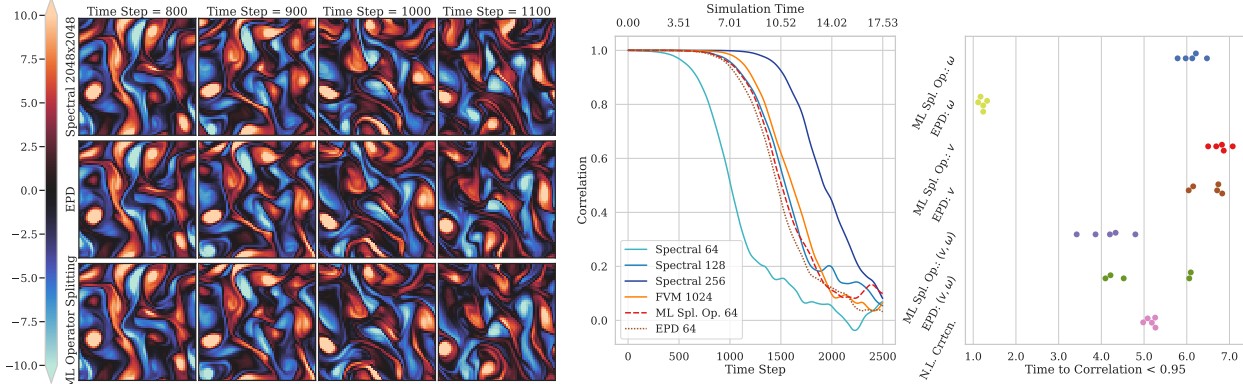

Figure 6: Benchmarking our Neural Split Operator (ML Spl. Op.) to spectral-only baselines and state-of-the-art ML-only, EPD model (Stachenfeld et al., 2022). *Left*: Visualizing model states starting at the time when divergence begins to occur. At time step = 800, EPD, Neural Split Operator model, and the high-resolution spectral-only 1024x1024 correlate well. By time step = 900, both learned models have begun to decorrelate with the ground truth albeit in different ways. *Right:* Comparing model variants across different random initializations. On the y-axis we measure the time to the first time step with correlation<0.95 with the high-resolution spectral ground truth. Then we compared our model with the EPD model across three data representation variations: velocity only ($\mathbf{v}$), vorticity only ($\omega$) and velocity-vorticity concatenation (($\mathbf{v}, \omega$)) as well as with an Nonlinear Term Correction model (N.L. Crrtn.). Shown here are the best performing models over five different random neural network parameter initializations.

Results are summarized in Figure 6. Comparing Spectral 128 to FVM 1024 we can see that the spectral-only method, without any machine learning, already comes close to a similar improvement. Thus, a ~2x improvement over the spectral-only baseline — achieved by both the EPD and our hybrid model — is comparable to the ~8x improvement over FVM achieved in Kochkov et al. (2021) in which they used a hybrid ML-Physics model.

**State representation.**   We compare three representations for inputs to neural networks in our EPD and split-operator methods: velocity, vorticity, and velocity-vorticity concatenation. All models represent internal state with vorticity and learn corrections to vorticity, with the exception of velocity-only EPD models. These models never compute vorticity — they use velocity to represent state as well as learned corrections, thus matching previous work by Kochkov et al. (2021) and Stachenfeld et al. (2022). Interestingly, this velocity-only representation has the best performance across the board, particularly for the fully learned EPD model, even outperforming velocity-vorticity concatenation (Fig. 6, right panel).

The nonlinear term $\mathbf{v} \cdot \nabla \omega$ in Equation (6) makes use of both vorticity and velocity representations. This indicates that in order to model the dynamics, the network must learn to solve for velocity — a global operation — making it challenging for a ConvNet restricted to local convolutions to learn. In contrast, computing vorticity from velocity only requires evaluating derivatives, which can be easily evaluated with local convolutions, e.g., via finite differences. This may explain the relatively-worse performance of Fourier Neural Operators using the velocity-based representation of Stachenfeld et al. (2022) versus the vorticity-based representation of Li et al. (2021).

**Full versus nonlinear-term-only correction.**   To understand the value of time-splitting for the learned correction, we also performed an experiment where we instead incorporated our ML model into the nonlinear part of Physics-Step (Eq. (3)). This entails solving the convection term with the same 4th order explicit Runge-Kutta method used in the numerical solver. Since velocity-space performed best with the split operator model, we also used it here. The non-split learned correction has significantly worse performance. Anecdotally, we observed that training with high order Runge-Kutta methods was significantly less stable. Models were much less robust to small changes in learning rates and dilation rates (App. C) than models trained with first-order Runge-Kutta. We believe this may be due to the increased difficulty of propagating

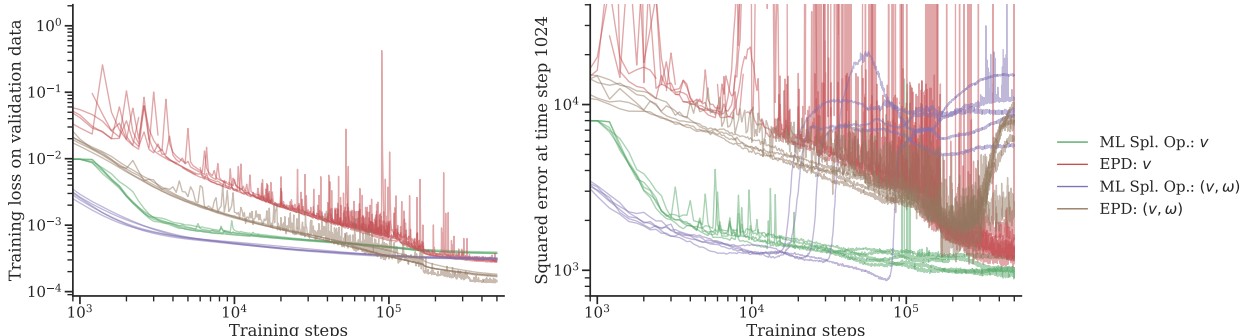

Figure 7: Learning curves for selected models, showing evaluation metrics after different numbers of training steps on the held-out validation dataset. The training loss is evaluated over the first 32 unrolled steps, whereas squared error is evaluated at 1024 steps ($t = 7.2$). The latter better indicates the model performance we care about for predicting long trajectories. Separate lines show learning curves for each of our five randomly initialized training runs.

gradients through a network which is effectively four times deeper. Another possibility is that the polynomial approximation of the high-order Runge-Kutta method introduces larger errors at the beginning of training when the model is less accurate.

**Analyzing learning curves.** To further compare the merits of different ML approaches, we compare the learning curves for select models in Figure 7. First, we notice that although our best EPD and hybrid models are similarly accurate once trained, the hybrid models require drastically less compute to achieve fixed evaluation metrics. For example, squared error of 2000 at 1024 time-steps requires only 3600 training steps for the median ML split operator velocity model, versus 164 600 training steps for the EPD velocity model, corresponding to 4.4 versus 170 TPU-core hours.

These learning curves also reveal that our validation loss, in this case calculated over 32 unrolled time-steps, is not necessarily indicative of validation performance over much longer unrolls. Although our models have never seen the exact validation data during training, many of them are still able to "overfit" to the task of predicting short trajectories, at the cost of generalization to long unrolls. We also observe that different ML architectures overfit to different extents. In particular, pure ML models and models with access to vorticity overfit more than our best hybrid model, the ML split operator with only velocity inputs. The ML split operator with access to both velocity and vorticity is particularly interesting, because it seems to undergo a phase transition at around 30 000 training steps, with correlation for long unrolls dropping dramatically as the model shifts into the "memorization" regime.

## 5 Discussion

In this paper, we demonstrated the potential of ML-augmented spectral solvers to improve upon the accuracy of spectral-only methods. We also identified several key physically motivated modeling choices — velocity-representation, first-order time stepping to improve sensitivity to hyperparameters, and the removal of global spatial artifacts — which improve training for both ML-only and hybrid models. Pure ML models can match the accuracy of hybrid models, but are considerably more expensive to train and show more indications of overfitting.

Traditional spectral methods are a powerful set of approaches for solving equations with smooth, periodic solutions. It is yet unclear whether ML-based solvers can achieve meaningful computational speed-ups over classical spectral methods on PDEs with smooth and periodic solutions. In contrast to prior work which showed computational speed-ups of up to 1-2 orders of magnitude over baseline finite volume (Kochkov et al., 2021) and finite difference methods (List et al., 2022), the roughly 2× decrease in grid resolution for 2D turbulence with the ML split operator would allow for at most 8× reduction in computational cost.

However, our neural network for learned corrections is about $10\times$ slower than Physics-Step, which counteracts this potential gain. Small speed-ups might be obtained by using smaller networks or applying corrections less frequently, but overall there is little potential for accelerating smooth, periodic 2D turbulence beyond traditional spectral solvers.

Hybrid ML-spectral methods may enjoy more significant improvements on other problems. Examples might include PDEs with less smooth solutions, such as 3D turbulence, where energy decays as $k^{-5/3}$ versus $k^{-3}$ in 2D (Pope, 2000). Or, global atmospheric models, where spectral methods do not achieve exponential convergence (Williamson, 2008). Cases where the exact governing equations are partially unknown, such as physical parameterizations for climate models (Brenowitz & Bretherton, 2018), also present an opportunity for combining physical simulation with machine learning components.

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
