# OpenReview forum: "Learning to correct spectral methods for simulating turbulent flows"
_TMLR — Rejected by TMLR_

### Review · Reviewer_6u4J · 2022-07-28

**Summary Of Contributions:**

Authors propose to use a neural network as a corrector (rescaled neural network output is added to the prediction for the next time step in a time-marching scheme) for classical pseudospectral solver and test their approach on three equations: Kuramoto-Sivashinsky, unstable Burgers, and 2D Navier-Stokes equation with Kolmogorov forcing. On these equations, authors benchmark against pseudospectral (Fourier) methods of various resolutions, the finite-volume method, and pure ML learned solver based on the linear encoder, decoder, and nonlinear dilated residual network. For considered equations, the proposed approach compares favorably with the competitors. However, computation-wise, it seems to provide no particular advantage compare with pure spectral solvers, as the authors note in the concluding remarks.

**Broader Impact Concerns:**

ok

**Requested Changes:**

Main concerns:

1. Classical subgrid modeling.

    It seems like the main contribution of the article is an ML way to replace subgrid modeling like RANS and LES. For example, it follows from the fragment: "Instead, coarse-grained approximations of the PDE are solved, known as “Large Eddy Simulation” (LES). LES augments Equation (1) by adding a correction term given by a closure model to account for averaging over fine spatial length scales. In practice, this term is often omitted due to the difficulty of deriving appropriate formulas (“implicit LES”), and the PDE is solved with whatever grid resolution is computationally feasible. In our case, correction terms are an opportunity for machine learning." But later, in all the experiments, this term is absent, and the result of the model corrected with the machine learning approach is compared with the results with no subgrid model. How do we know that the learned ML subgrid model is better than the classical models? The reviewer believes that comparison (or at least extended discussion) with some classical nontrivial closure model will significantly improve the quality of the present article. The suggestion is either to perform the additional experiment with the classical closure model or to include a discussion.
2. The choice of baselines for PDEs.

    The choice of baselines seems arbitrary and is explained nowhere. Namely, it is not clear why for the first two equations authors decided not to include ML and FEM baseline. This warrants some explanation.
3. Mentioned metrics are not provided for all PDEs.

    As performance measures, the authors use the correlation coefficient and mean absolute error. However, these measures are not computed for all considered equations. For the KS equation, both metrics are computed, for the Burger equation only mean absolute error is computed, and for Navier-Stokes, only the correlation coefficient is computed. The reviewer suggests reporting all performance measures either in the main text or in the supplementary materials.
4. Stability.

    "In contrast to prior work which showed computational speed-ups of up to 1-2 orders of magnitude over baseline finite volume (Kochkov et al., 2021) and finite difference methods (List et al., 2022), the roughly 2x decrease in grid resolution for 2D turbulence with the ML split operator would only allow for at most 8x reduction in computational cost. However, our neural network for learned corrections is about 10◊ slower than Physics-Step, which entirely counteracts this potential gain. Small speed-ups might be obtained by using smaller networks or applying corrections less frequently, but overall there is little potential for accelerating 2D turbulence beyond traditional spectral solvers." One of the bottlenecks for spectral methods is the CFL condition. It is an interesting question whether ML can stretch the stability region of the spectral method. The reviewer suggests including additional experiments to test the hybrid ML model for the longer-that-allowed time steps. As other publications indicate, ML approaches often allow for improved stability region.

Minor points (occasional inconsistencies in the terminology, unclear parts, questionable statements):
1. The line in the abstract "A recent line of work indicates that a hybrid of classical numerical techniques with machine learning can offer significant improvements over either approach alone" implicitly suggests that the hybrid approach advocated in the article is superior to classical methods. The conclusion written by the authors suggests otherwise. The reviewer proposes to correct this inconsistency and explicitly explain in the abstract that the proposed neural network is not numerically attractive.
2. "We leverage the physical intuition that information travels at finite speed — for a small time step h, the solution at time t + h only depends locally in space on the solution at time t (within a dependency cone usually encoded by the Courant–Friedrichs–Lewy (CFL) condition)." It is better to reformulate this sentence somehow. That information travels at finite speed is known from physics and has nothing to do with grid spacing. On the other hand CFL condition involves grid spacing but is not directly related to physics.
3. "an implicit Crank-Nicolson method" Crank-Nicolson method is by definition implicit.
4. "The most suitable representation for a given numerical solver may not necessarily be the most suitable representation for ML." For ML solver/ML method?
5. Figure 3 contains a few unclear elements.

    a. Velocity-Solve is used presumably to denote the transformation from vorticity to velocity. In the scheme, the same element is called State-Transform. For example, "... but for 2D Navier-Stokes (depicted in Fig. 3) we use State-Transform to calculate velocity from vorticity." One should use either Velocity-Solve or State-Transform but not both.

    b. It is stated in the text that "Thus, in addition to filtering the downsampled training data, we used the same exponential filter on the outputs of Physics-Step of Equation (3)." Does the final scheme contain this filter or not? It is not shown in figure 3.
6. "Shock waves are notoriously difficult to model using the Fourier basis due to the discontinuous solution (Sec. 3.4). Due to the mixing of length scales introduced by convection, these errors can quickly propagate to affect the overall dynamics. This can result in instability at low resolutions, evidenced in our results by an unstable coarse model (resolution 32). The inadequacy of spectral methods for solving this problem also explains why our method is able to achieve approximately 4x improvement in spatial resolution over a spectral baseline." This part is not completely fair. There are spectral methods for problems with shock waves. Approaches with spectral viscosity provide such an example. Should clarify at least that _the standard_ spectral methods can not resolve shocks.
7. What is the "Nonlinear Term Correction model"? Presumably, this is a model that is used as a correction that is integrated by the RK4 method. The precise meaning of "N.L. Crrtn." should be explained somewhere.
8. Figure 6. The colors on the last (on the left) graph on this figure are misleading. As the reviewer understands, the name of a model appears on the y-axis in black color. On the other hand, the graph on the left uses similar colors for different models.

**Strengths And Weaknesses:**

Strong points

1. Paper is well-written.
2. Necessary background from spectral methods is introduced.
3. A diverse set of solvers is used for benchmarks.
4. Discussion of results is impartial and realistic.

Weak points (see section Requested Changes for a detailed description)

1. Classical subgrid modeling is mentioned but is absent from benchmarks.
2. The choice of baselines for PDEs is not justified.
3. Mentioned metrics are not provided for all PDEs.
4. Occasional inconsistencies in the terminology, unclear parts, questionable statements.

---

### Review · Reviewer_ziVT · 2022-08-12

**Summary Of Contributions:**

This paper proposes a hybrid ML method for solving PDEs governing turbulent fluid flow. In particular, this paper proposes to combine the spectral method, a physical step with finite Fourier basis representation, and a neural network, a learned correction with CNN architecture, in deriving the hybrid solution, as prescribed in Equation $(3)$ the split operator.

On top of the proposal of the framework, this paper also presents the motivation of using spectral method, the design idea of balancing global and local features, and nuances to support/facilitate the hybrid.

Authors conduct empirical study on two 1-D problems, for proof-of-concept, and one 2-D problem to demonstrate the practical performance.

**Broader Impact Concerns:**

I do not see any particular concerns in this work.

**Requested Changes:**

I'd like to highlight a few changes/questions on the presentation of the results, especially on the figures.

$\textbf{1)}$. The order of figures could be re-arranged. E.g., Figures 3 and 5 were mentioned in the text prior to Figure 2 etc.

$\textbf{2)}$. It might be ok not to present an architecture of NN as it is based on EPD, but for context, it might be beneficial to display a detailed architecture.

$\textbf{3)}$. The displayed measures are not very consistent, e.g., Figure 4 vs Figure 5, is correlation less important for Figure 5, so it was not shown? Training loss were not displayed for toy problems.

$\textbf{4})$ Could there be any 3-D problems included in the experiments? Or it is not applicable/necessary?

**Strengths And Weaknesses:**

$\textbf{Strength}$

This paper is a good addition to the existing hybrid ML method for turbulent flow domain. The novelty is on proposing to combine spectral method and neural network, in particular, to derive a learned correction and to compensate spectral method by capturing local features. This paper is very well organized with comprehensive literature, illustrations on motivations, and very well detailed explanations on the proposed approach. I found the numerical study is especially comprehensive in terms of data preparation, setup, training steps and so on, and these types of technical details are often neglected by other literature (but details that are really helpful for reproduction).

$\textbf{Weakness}$

First of all, the weakness in novelty is highlighted by using known approaches as the components in the proposed method. The network is basically a known CNN architecture, Encoder-Process-Decoder (EPD). I am not sure how significant the contribution would be in terms of a machine learning solution.

Second, the empirical study only covers 1-D and 2-D problems, and the spectral method is known to work very well for smooth function but not necessarily in general. The general applicability of the proposed approach is yet to be determined. It seems that authors also highlight this in the discussion chapter.

---

### Review · Reviewer_ZMNQ · 2022-08-15

**Summary Of Contributions:**

The authors bring together Fourier-based spectral methods with machine learning techniques in order to solve partial differential equations (PDEs). The main idea of the proposed method is to learn appropriate corrections to spectral methods via simulations.

The authors explore experimentally the proposed method, both in situations where the proposed method can provide substantial improvements over current methods, as well as on a 2D turbulence task where it is shown again that the hybrid approach of spectral methods with machine learning, provide better results compared to spectral-only methods (which already provide very good results compared to the current state of the art).

**Broader Impact Concerns:**

The authors should add a clear section discussing the implications and concerns on the broader impact of their work.

**Requested Changes:**

As I mentioned above I am not capable of appreciating fully the contributions of the paper as it is way outside my area of expertise. Judging from machine learning papers that I usually read, the presentation of this paper is very bad and for this reason in this current form I can only suggest a major revision by addressing the following presentation points. The opinion of reviewers who have knowledge on physics-based methods and anything non-machine learning related to this paper, should be seriously taken into account so that it can be understood where this paper stands with respect to the main axes that are driving it. Below are my comments for improving the presentation.

Requesting explanations:
- page 1: "... have achieved improved stability and generalization in addition to accuracy"
I have no idea what the authors are trying to say. In machine learning improved generalization means improved accuracy.
- page 4: "... errors decay super-algebraically ..." -> explain what this means.


Figures should appear after they are mentioned in the text.
- Figure 1 appears in page 3 and is mentioned at the end of section 2 in page 4.
- Figure 2 appears in page 4 and is referenced in page 6 (paragraph 3), even after Figure 3 is mentioned in the text!
- Figure 3 appears in page 5 and is referenced in page 6 (paragraph 1) before Figure 2 is referenced
- Figures 4 and 5 are mentioned in page 6 (paragraph 3) before Figure 2.

What kind of story do you want to tell?  You need text for the figures that appear in the paper _before_ the figures actually appear. Please explain the figures beyond their caption and in fact, before you present them to the reader.


Typos:
- page 1: unfeasible -> infeasible
- page 3: equation 2: in the vector of K+1 frequencies, shouldn't the subscript be 0 in the middle instead of k?
- page 4: Fast Fourier transform (FFT) -> FFT is in textsuperscript instead of just plain capital letters.
- page 5: last word of section 3.1: "detail" -> details.
- page 5: first sentence of section 3.2: there are two occurrences of the word "different". Are you sure the second occurrence is the word that you wanted to have in that sentence?
- page 5: last sentence: "... serves as strong baselines ..." -> "serves" is singular and "baselines" is plural.
- page 5: paragraph "Training loss." -> when you write l2-loss, you mean square loss, right? (It could be an l2-norm the way it is written.) Perhaps some preliminaries would help the paper where you introduce notation and terminology?


**Strengths And Weaknesses:**

Strengths:

The paper proposes a physics-based machine learning method in order to provide better solutions.  I consider this an important step on the applications of machine learning, as the resulting solutions have to respect certain constraints that are imposed by various physical models.

Weaknesses:

I think the paper is hardly self-contained for anyone who has a background in machine learning only - which is the situation about myself.  I can understand where the authors attempt to innovate but I have no idea what the state of the art is and if the claims that the authors make in terms of models that are related to physics / PDEs are correct/accurate.

The presentation of the paper could improve, please see below for requested changes.

---

### Author Response · Authors · 2022-08-22
**Response to Reviewers**

We thank the reviewers for their careful reading, along with their thoughtful and detailed feedback. We agree with all the minor changes and typos and will make changes accordingly. We will fix the order of the figures [ziVT, 6u4J].

Reviewer 6u4J points out that our abstract “implicitly suggests that the hybrid approach advocated in the article is superior to classical methods.” We will make sure to clarify this in the final version.

ZMNQ recognizes the importance of this work and defers the physics content to the other reviews [ziVT, 6u4J] who agree that overall, the paper is well-organized and clear.

We address the high level feedback below.

# Mean Absolute Error (MAE) vs. Correlation [ziVT, 6u4J]

Generally, we prefer to report correlation because it is more interpretable. However, we also report MAE when it provides extra information. In our experiments involving the 2D Navier-Stokes equation, we only reported correlation because MAE did not provide any additional insights and was therefore omitted for concision. In our experiment involving the unstable Burgers’ equation, the situation was the opposite: We only reported MAE because measuring correlation did not provide extra insight — all models had correlation values close to 1.0, whereas MAE was able to measure the improved performance of the method. For the KS equation, we included both MAE and correlation to demonstrate the redundancy between these two metrics. We will add this explanation in the “Measuring convergence” paragraph before Section 4 (pg. 7).

# Choice of baselines [6u4J]

We chose equations whose solutions exhibit different typical behaviors in order to showcase the behavior of our proposed model in controlled settings. The experiment involving the KS equation demonstrates that our method does improve, though not significantly, on a smooth, chaotic problem where classical spectral methods excel. The unstable Burgers’ equation has pseudo-singularities in which  our model performs significantly better. The 2D Navier-Stokes equation has solutions which exhibit multiscale behavior in addition to smooth, chaotic dynamics. Ultimately, the comparisons to the other baselines for the 2D Navier-Stokes equation are identical to the 1D equations. We will clarify this in “Section 4.1 Baselines.”

# Classical subgrid modeling [6u4j]

Classical subgrid modeling, such as Large Eddy Simulation (LES), is itself a nuanced class of methods with many tunable parameters. Implicit LES is a widely-used and well-understood model which serves as a consistent and parameter-free baseline. In contrast, explicit sub-grid-scale models such as Smagorinsky models include tunable parameters, with the optimal choice depending on the scenario of interest. Furthermore, explicit LES models typically focus on matching the energy spectrum rather than minimizing point-wise errors. Both of these points are confirmed in a [recent study](https://arxiv.org/pdf/2202.06988.pdf) [1]. We will include an explanation similar to this in Section 4.1 “Baselines.”


[1] List, Björn, Li-Wei Chen, and Nils Thuerey. "Learned Turbulence Modelling with Differentiable Fluid Solvers." arXiv preprint arXiv:2202.06988 (2022).

---

> ### Author Response · Authors · 2022-09-14
> **Updated manuscript in progress**
>
> Dear reviewers,
>
> We are working on an updated manuscript which will be uploaded in about one week, if not sooner. This manuscript will address all of your comments.
>
> Thank you for your patience and best wishes.

---

### Decision · Action_Editors · 2022-10-30

**Recommendation:** Reject

**Comment:**

The paper proposes an "Neuro-Split" approach. It combines a physical time-stepping scheme with learned correction. The solution is presented in a spectral basis, compared to finite volume methods used in the previous papers.

The reviewers note that the results are interesting, but questions about baselines and metrics have been raised.

I recommend to reject this paper. The main reason for the rejection is that the authors at some point submitted the revision which lists all the authors, breaking the anonymity of the submission. Also, If the authors decide to resubmit, they need not to refer to this OpenReview page in order not to break the anonymity again.

Lets me also stress other important concerns.

1) The text is focused more on physics and numerical methods, rather on machine learning techniques. The overall idea is quite simple: collect a dataset using well-developed numerical solver and learn a CNN to predict the correction term. The innovation compared to previous works is the spectral solver and the creation of the dataset on the coarse grid by subsampling and applying the filter.

2) The correct comparison should include comparison to other models, mentioned by the reviewers. The most important is to include the comparison with mentioned subgrid modelling techniques. The authors in their answer said that it is possible, but different metrics are used, but this does not give an understanding how the method will performs.



**Audience:**

There is quite a lot of research on applying ML for physical problems, so quite a lot of people could be interested.

**Claims And Evidence:**

The claims of the paper are supported by numerical evidence on several standard examples.
However, some baselines (such as more efficient classical models) and metrics (such as energy spectrum standard for subgrid models and more important for turbulence modelling) are not computed.